# Neuroendocrine Aspects of Skin Aging

**DOI:** 10.3390/ijms20112798

**Published:** 2019-06-07

**Authors:** Georgeta Bocheva, Radomir M. Slominski, Andrzej T. Slominski

**Affiliations:** 1Department of Pharmacology and Toxicology, Medical University of Sofia, Sofia 1431, Bulgaria; 2Division of Rheumatology, Department of Medicine, University of Alabama at Birmingham, Birmingham, AL 35294, USA; radomir.slominski@gmail.com; 3Department of Dermatology, Comprehensive Cancer Center, Cancer Chemoprevention Program, University of Alabama at Birmingham, Birmingham, AL 35294, USA; 4Veteran Administration Medical Center, Birmingham, AL 35294, USA

**Keywords:** skin aging, photoaging, premature aged skin, UV irradiation, oxidative stress, vitamins B_3_ and D, melatonin

## Abstract

Skin aging is accompanied by a gradual loss of function, physiological integrity and the ability to cope with internal and external stressors. This is secondary to a combination of complex biological processes influenced by constitutive and environmental factors or by local and systemic pathologies. Skin aging and its phenotypic presentation are dependent on constitutive (genetic) and systemic factors. It can be accelerated by environmental stressors, such as ultraviolet radiation, pollutants and microbial insults. The skin’s functions and its abilities to cope with external stressors are regulated by the cutaneous neuroendocrine systems encompassing the regulated and coordinated production of neuropeptides, neurohormones, neurotransmitters and hormones, including steroids and secosteroids. These will induce/stimulate downstream signaling through activation of corresponding receptors. These pathways and corresponding coordinated responses to the stressors decay with age or undergo pathological malfunctions. This affects the overall skin phenotype and epidermal, dermal, hypodermal and adnexal functions. We propose that skin aging can be attenuated or its phenotypic presentation reversed by the topical use of selected factors with local neurohormonal activities targeting specific receptors or enzymes. Some of our favorite factors include melatonin and its metabolites, noncalcemic secosteroids and lumisterol derivatives, because of their low toxicity and their desirable local phenotypic effects.

## 1. Introduction

The skin is a complex multifunctional self-regulating organ in the human body. Its functions are critical to survival. The skin is not only a barrier that protects the organism from the deleterious insults of the external environment, but it is also crucial for thermoregulation, as well as its maintenance of electrolyte and fluid balance. Moreover, the skin also responds to environmental changes, such as biological, chemical, and physical factors, in order to regulate cutaneous and global body homeostasis [1,2,3].

It is well established that in the skin there is an important sophisticated network connecting cutaneous nerves and the local neuroendocrine and immune systems. The brain directly (via efferent nerves) or indirectly (via the adrenal glands or immune cells) regulates skin function. The neurocutaneous communication comprises of afferent and efferent nerves that release mediators acting on corresponding receptors expressed on skin cells [1,4]. Furthermore, as a sensory organ with neuroendocrine activities, the skin can also transmit humoral or neuronal signals to the central nervous, endocrine and immune systems. In addition, environmental factors or pathological processes induce skin changes that can imprint circulating immune cells acting as cellular messengers of skin responses to the changes in local homeostasis [1,2]. The skin also operates as a biofactory for the synthesis, processing and metabolism of the wide range of structural proteins, glycans, lipids and signaling molecules [5], as well as a fully functional neuroendocrine organ [6,7]. The human skin produces a variety of hormones, neuropeptides and neurotransmitters [1,2,3,8] in addition to the formation of vitamin D3 [9,10,11]. The skin responds to stress (such as UV light) by local synthesis of all hormones of the classical hypothalamic-pituitary-adrenal (HPA) axis [12]. Specifically, skin cells are capable of producing corticotropin-releasing hormone (CRH) [13,14,15,16,17,18,19,20,21], CRH-related peptides including urocortin 1 and 2 [3,22], proopiomelanocortin (POMC)-derived ACTH, α-MSH and β-endorphin [3,13,23,24,25,26,27,28], and glucocorticoids [29,30]. They also express the corresponding receptors. There are also many other hormones synthetized or activated/inactivated in the skin, including thyroid releasing hormone (TRH), thyroid stimulating hormone (TSH) and thyroid hormones, [31,32,33,34]; sex hormones and their precursors, as well as ∆7 steroids and different secosteroidal products [7,29,35,36,37,38]. The skin expresses the enzyme cytochrome P450scc (CYP11A1), which initiates steroid synthesis by converting cholesterol to pregnenolone in a similar manner as in other steroidogenic tissues [36,38,39,40,41,42,43,44,45]. In addition, skin cells can produce catecholamines [46,47], serotonin [48,49,50,51], and melatonin [48,50,52,53,54,55]. Indeed, melatonin and its biologically active metabolites are essential for physiological skin functions and protection against environmental stress [48,54,55,56,57,58].

## 2. Skin Aging

Aging is a natural process leading to the accumulation of damage and progressive deterioration in the biochemical, physiological and morphological functions on the systemic or organ levels [59,60]. Chronobiological aging mainly results from imbalanced endocrine circadian rhythmicity, which is linked to numerous health complications and pathologies in aging populations. Many factors can cause or aggravate hormone deficiencies (like nutritional, dietary, lifestyle, behavioral, environmental deficiencies, etc.) [61,62]. These hormonal changes induce morphological and functional alterations of all organs and systems, including the central nervous system (CNS )and skin. Moreover, the physiological aging process results in most of the phenotypic changes observed in the skin. There are age-related changes affecting all endocrine glands, which sometimes are so intertwined that the reduced function in one gland affects the other one [2,7,63]. Aging affects the expression of POMC and production of POMC-derived peptides, especially of melanocortin receptor 1 (MC1R) and MC2R agonists, which are of crucial importance for skin biological systems [2,64]. The regulation of the skin steroidogenic system cannot be underestimated, since it can regulate epidermal functions and skin immunity [7,38]. The breakdown of this steroidogenic activity can lead to pathological skin changes and diseases. The abnormal synthesis of skin cholesterol, involving a drastic reduction in steroids, is associated with down-regulation of epidermal differentiation [7,38,65]. Furthermore, the levels of steroidogenic acute regulatory protein (StAR) mRNA were found to gradually decrease in the skin tissues of elderly people, in contrast to younger ones [66]. With increasing age, the capacity of the skin to produce vitamin D3 declines, thus its protective effects are reduced [67,68]. Several factors contribute to this vitamin D deficiency state, such as behavior factors (limited sun exposure, malnutrition, etc.) and reduced synthetic capacity [69].

## 3. Factors Affecting Skin Aging

The skin, like all organs, follows the pathway of aging accompanied by a gradual loss of cellular functions and physiological integrity, and is a mirror of the first signs of aging [70]. Skin aging is a complex biological process influenced by internal (constitutive) and external (environmental) factors, leading to cumulative changes of skin structure, function and appearance [71]. Skin aging can be classified as physiological (chronological) aging and environmentally-induced, including photoaging.

The internal factors influencing chronological aging affect all skin areas and include genetic (changes in gene expression), changes in the neuroendocrine system (e.g. physiologic decline of hormones), development of skin disorders disrupting the cutaneous barrier functions or skin involvement in connective tissue disorders. The rate of aging can vary among different populations determined by differences in anatomy and physiology, as well as among different anatomical areas even within the same individual [72]. High levels of pigmentation form a natural protective shield against UV radiation. The pigmentation level of the skin is genetically determined by the type, distribution and density of melanin pigment, which can transform the absorbed UV radiation into heat, thereby reducing UV-induced cell damage and subsequent skin aging [73]. Interestingly, there is evidence that the pigmentation-related MC1R gene encoding the corresponding MC1R receptor is a key regulator of eumelanogenesis. Diminished MC1R activity due to loss-of-function leads to the production of pheomelanin, which has a weaker UV shielding capacity than that of eumelanin [74]. The DNA variants in MC1R are significantly associated with perceived facial age, providing a new molecular base for youthful looks [75].

In addition, several environmental factors accelerate the onset of aging in the skin, leading to premature skin aging (Figure 1).

The external factors affect areas of the body especially exposed to the environment, such as the face, head, neck, and hands. The main external factors are ultraviolet radiation (UVR) [71,76], tobacco smoking [77,78], and other environmental pollutants and toxins [79,80].

UVR can regulate global homeostasis after absorption and transduction of its electromagnetic energy into chemical, hormonal, and neuronal signals [81]. This homeostatic activity includes activation of the central neuroendocrine pathways [81]. Sun exposure not only has benefits, but also risks as well. Chronic exposure to UVR is the most harmful environmental factor affecting skin biology according to the anatomic location and skin type. It leads to premature skin aging, a process also known as photoaging [82]. Solar UVR that reaches the earth’s surface has wavelengths ranging from 280 to 400 nm, divided into UVA (320–400 nm) and UVB (280–320 nm). Exposure to UVB has a larger biological impact on the skin than that of UVA at similar radiation doses. UVB does not penetrate deeply into the skin and is largely responsible for the development of sunburn. UVA has better penetration and reaches the reticular dermis, but is 1000 times less efficient in induction of biological effects (e.g. minimal erythema dose) compared to UVB [83]. Several studies have shown that long-term exposure to UVA can damage the dermis more significantly than UVB, leading to photoaging and free radical production. The UVA/UVB ratio is approximately 10/1 with the sun in the overhead position. The radiation intensity of both UVA and UVB depends on many parameters, including latitude, season, time of the day, meteorological conditions and ozone layer [84]. The dose of radiation presented in J/m^2^ represents the radiation intensity multiplied by the exposure time. It determines the magnitude of UVR-induced skin damage. In comparison to indoor-workers, outdoor-workers accumulate a higher total UV dose, being therefore at higher risks of development of premature skin aging and skin cancers such as basal and squamous cell carcinomas [85,86], but they are at lower risk of developing melanomas. Melanomas are seen mainly in indoor-workers and are associated with intense intermittent exposure and developing of sunburns [86]. Most biologically relevant chromophores absorb UVB and UVC (from artificial UVC-sources). In contrast, UVA is weakly absorbed by DNA and by limited cellular chromophores, but induces oxidative damages [81].

Although UVR causes photoaging, environmental pollutants can also damage the skin. Air pollutants such as noxious gases, together with UVA, can act synergistically in initiation of skin cancers. In addition, particulate matter (PM) pollutants induce skin aging through penetration of the epidermal layer of the skin and through adnexal structures [80,87]. In addition, new evidence suggests that environmental pollution, particularly persistent organic pollutants (POPs), can interfere with the endocrine system by behaving like endocrine-disrupting chemicals (EDCs). EDCs can affect the biosynthestic pathways of steroid and thyroid hormones and their systemic levels [88,89]. Air pollutants, especially ozone and PM can directly affect the cutaneous production of vitamin D. Furthermore, EDCs may inhibit the activity and expression of Cytochrome P450 (CYP) and indirectly can cause vitamin D deficiency through weight gain and dysregulation of the thyroid hormone, parathyroid hormone, and calcium homeostasis. In addition, smoking can lead to a decrease in serum levels of 25(OH)D_3_ and 1,25(OH)_2_D_3_ [79]. Miscellaneous lifestyle components such as diet, sleeping position and overall health also affect the appearance of the skin [72].

## 4. Skin Structure and Morphological Changes in Advanced Age

As a multi-layered organ, human skin comprises of external, stratified, non-vascularized epidermis, underlying connective tissue (dermis), subcutaneous adipose tissue defined as hypodermis, and adnexal structures [90]. The epidermis is predominantly composed of self-renewing keratinocytes, which generate solid lipid-rich cornified layers during differentiation [91]. Stem cells (SCs), located in the basal layer, and transient-amplifying (TA) cells are important for epidermal regeneration. Dysregulation of keratinocyte SCs may result in skin aging [92,93]. Epidermal melanocytes produce and transfer melanin pigment to keratinocytes as an important element of skin protection against UVR damage [94]. Basement membrane separates the epidermis from the dermis, restricting communication between these components.

The dermis consists principally of fibroblasts/fibrocytes, which are mesenchymal cell types producing fibrous and elastic components responsible for cutaneous strength and elasticity, as well as proteoglycans (PGs), glycoproteins, water and hyaluronic acid (HA), and other biologically active molecules, together called the extracellular matrix (ECM) [90]. HA as one of the glycosaminoglycans (GAGs) forms proteoglycan aggregates which crosslink to other matrix proteins such as the collagen network, leading to an increase in tissue stiffness [95]. In contrast to keratinocytes, the resident cells and the fibers of the dermis have lower regenerative ability. Moreover, in vitro studies show that human fibroblasts are more susceptible to UV exposure than the epidermal keratinocytes [96]. This may have implications in vivo only for sun radiation with wavelengths able to cross the epidermis and reach fibroblasts, that is, >310 nm for phototype I or >340 nm for phototype V [97,98].

Additional skin components are the immune cells, including lymphocytes, macrophages, mast and dendritic cells. They predominantly reside in the dermis but sometimes they are present in the hypodermis as well. The hypodermis is important for energy storage. The adnexa are located in both the dermis and the hypodermis depending on their activities and functions. Human adnexal structures include hair follicles, sebaceous glands, eccrine glands, and apocrine glands. All structures in the skin are supplied by a network of somatosensory and autonomic nerve fibers, as well as by vascular and lymphatic networks [1].

With accelerating age, skin functions deteriorate due to structural and morphological changes. Also, the cutaneous regenerative potential declines with age. Keratinocyte SCs and fibroblasts undergo senescence and the accumulation of such senescent cells over time reduces skin regeneration capabilities, contributing to skin aging [93,99].

Endogenous aging of the skin is mainly influenced by genetic and metabolic factors acting in an age-dependent fashion. Skin at advanced ages is characterized by 10–50% of epidermal thinning, fragility, fine wrinkle formation, and loss of elasticity [71]. The thinning of the epidermis depends on progressive dysfunction of keratinocytes with SC-like properties and lower epidermal turnover, which are associated with a decline of skin barrier functions and capability of wound healing [70,71]. It is assumed that the chronologically aged skin is intrinsically less hydrated, less elastic, more permeable and susceptible to irritation. The chronological dermal remodeling is mainly due to dysfunction of long-lasting resident fibrocytes that constantly undergo damage accumulation [100]. Senescent fibroblasts lose the ability to organize the ECM by reduction of collagens and elastins synthesis.

The histological features of aged skin are epidermal atrophy (atrophy of stratum spinosum), flattening of dermoepidermal junction, reduction of dermal thickness and atrophy of ECM, reduction of adnexal structures and decrease of their functions, thinning of subcutaneous fat, and reduction in the number of nerve endings and cutaneous microvessels. There is also increased heterogeneity in the size of basal cells, which often show decreased mitotic activity. There is a decrease in number of melanocytes and Langerhans cells, and in number of dermal fibroblasts. Collagen and elastic fibers are thin, loose, and disintegrated [70].

## 5. Morphological Changes in Prematurely Aged Skin

Skin damage due to chronic sunlight exposure accounts for up to 90% of visible skin aging, in particular on the face of people with a light complexion (skin types I and II) [72]. Photoaging is the superposition of the solar damage on the normal aging process resulting in premature skin aging. The clinical signs of photoaging include deep wrinkles, skin laxity, early appearance of lentigines and dyschromia, sallow yellow color, loss of normal translucency and gradual appearance of telangiectasia (Figure 2). While the primary effects of photodamage include epidermal thickening, additional photodamage can lead to significant thinning of the skin [72,101]. Atrophy and chronic skin fragility, senile purpura, and pseudoscars are morphological signs of dermatoporosis [102] seen mainly around 70 years of age [103,104].

Another important external factor leading to premature skin aging is smoking. Smoking increases keratinocytic dysplasia and roughness of the skin and a dose-dependent relationship between wrinkling and smoking was found [72,78]. According to some authors, smoking is considered to be a greater contributor to facial wrinkling than the sun exposure [105].

Histological features characterizing prematurely aged skin include epidermal thickness heterogeneity (thickening in the beginning, then thinning), pleomorphic corneocytes in sun-exposed areas, flattening of the dermoepidermal junction, increased number of mast cells and neutrophils, stellate phenotype of fibroblasts, and extensive damage of dermal connective tissue (solar elastosis), which is a hallmark of photoaged skin. Major alterations occur primarily in the dermis, resulting in degeneration of collagen, deposition of abnormal elastic material, increased level of dysfunctional GAGs and PGs, and dilated vessels with thickened walls [70].

The synergic effects of environmental and internal aging factors over the human lifespan impair the cutaneous barrier function with significant morbidity [101]. Aged skin is susceptible to pervasive dryness and itching, cutaneous infectious diseases, autoimmune skin disorders, vascular complications (telangiectasia, senile purpura, etc.), senile lentigines and other pigmentory changes, and so on. Other age-associated skin diseases include benign skin changes, such as seborrheic keratosis, premalignant lesions of solar keratosis and lentigo maligna, as well as melanoma and non-melanoma skin cancer [70,72].

## 6. Molecular Mechanisms of Skin Aging

Understanding the molecular mechanisms of skin aging is of great importance to create a preventative anti-aging strategy, to delay the onset of aging, and to reduce the age-associated skin damages and diseases. Changes in gene expression, generation of reactive oxygen species (ROS) by oxidative metabolism, decreased antioxidant defense, telomere attrition, and defects in cellular DNA repair form the basis for chronological aging. The replicative abilities of keratinocytes, fibroblasts and melanocytes decrease with time, leading to senescent, non-dividing cells. p16^INK4a^ and p63 (p53-related protein) are mediators of keratinocyte senescence. Specifically, p16^INK4a^ expression correlates with chronological aging of human skin in vivo. Moreover, the number of p16^INK4a^-positive cells in both epidermis and dermis increase with age [106]. In contrast, aged keratinocytes show reduced expression of p63 [107]. In particular, p63 deficiency in adult mice causes a cell growth arrest and induces appearance of aging features [108].

In addition, in human dermal fibroblasts, sirtuin (SIRT)-1 expression is significantly reduced in advanced age [100]. SIRT 1–7 belong to a family of nicotinamide adenine dinucleotide (NAD)-dependent histone deacetylases. SIRT1, SIRT3, and SIRT5 can protect the cell from ROS, while SIRT2, SIRT6, and SIRT7 can modulate crucial oxidative stress response mechanisms [109,110,111,112]. SIRT1-up-regulation or down-regulation results in delayed or accelerated fibroblast senescence, respectively [113]. Similar to SIRT1, SIRT6 is implicated in aging, but it modulates the accessibility of DNA repair proteins to chromatin [114]. Epigenetic mechanisms also mark cell senescence and epigenome modifications contribute to the aging process [93].

Accumulating evidence supports a strong link between mitochondrial dysfunction and aging [115,116]. Many reports suggest a decrease in mtDNA content and mitochondrial number with advancing age [117,118].

In the skin, approximately 1.5–5% of the oxygen consumed is converted into ROS by intrinsic processes [119]. These ROS can trigger a degradation of dermal ECM. The photoaging is primary due to chronic exposure to UVR, which, by damaging multiple cellular structures, accelerates the aging process. UVA exposure increases the expression of proteolytic enzymes (such as matrix metalloproteinases) resulting in disorganization and progressive degeneration of the ECM [120]. Chronic UVA irradiation inhibits hyaluronan synthesis via down-regulation of the hyaluronic acid synthases (HAS)-1, -2, -3, thus altering the composition of PGs [100]. In addition, photo-aged fibroblasts with senescent phenotype increase melanogenic gene transcription, causing hyperpigmentation and appearance of “senile lentigines” [121].

UVB radiation, absorbed mainly by epidermal DNA and RNA, can lead to various mutations, including so called “solar UV signature” and production of dysfunctional proteins. The first UVB chromophore encountered by UVB radiation penetrating the skin is *trans*-urocanic acid (UCA), which is an endogenous sunscreen with low level protection against DNA damage and apoptosis [122]. However, *trans*-UCA undergoes a *cis*-*trans* isomerization to *cis*-UCA, which is believed to mediate, at least in part, UVB-induced immunosuppression [123]. An accumulation of unrepaired mutations can cause cycle arrest or apoptosis, or lead to carcinogenesis [76]. Although some aging mechanisms share several similarities or overlaps, photoaged skin and chronically aged skin show different changes in the ECM. Photoaged skin is characterized by damaged collagen and accumulated aberrant elastin fibers and GAGs, whereas endogenous aged skin shows atrophy of the dermal structures [124].

Generally, skin aging is mainly initiated by oxidative events. In particular, extensive ROS production due to insufficient scavenging activity or an altered mitochondrial function is crucial in oxidative stress-induced skin aging [125]. As a consequence of the oxidative stress, high levels of ROS lead to oxidative damage of lipids, proteins, genomic DNA, mitochondrial DNA (mtDNA), and also can deplete and damage non-enzymatic and enzymatic antioxidant defense systems of the skin. An important target for ROS is mtDNA and its damage and decline in function lead to vicious cycle-like effects, resulting in enhanced ROS production (see Figure 3) [125]. Accumulation of ROS dysregulates cell signaling pathways alters cytokine release and leads to inflammatory responses [126].

The aging process includes the activation of nuclear factor-κβ (NF-κβ) and activator protein-1 (AP-1), which are redox sensitive transcription factors involved in inflammation and wrinkle formation. [127]. Both transcription factor complexes are elevated within hours of low-dose UVB irradiation of the skin. Increased levels of ROS induce activation of mitogen-activated protein kinases (MAPKs) such as extracellular signal-regulated kinases (ERK), MAPK p38, and transcription factor c-Jun-N-terminal kinase (JNK) in the AP-1 pathway. In addition, upstream signaling enzymes (inhibitor of κBα, AKT-Protein kinase B, etc.) in the NF-κβ pathway are upregulated. Normally, ERK mediates cellular responses to growth factors, whereas JNK and p38 mediate cellular responses related to cytokines and physical stress [127,128,129]. Activation of ERK and p38 results in the degradation of ECM and down-regulation of neocollagenesis [129].

NF-κβ signaling is a well-known regulator of tissue homeostasis. Recently, its central role in skin aging was underlined [130]. Thus, NF-κβ had increased expression in mtDNA-depleter mice, and after restoration of mtDNA, the NF-κβ expression was reduced. These data confirm that NF-κβ signaling is a critical mechanism contributing to skin and hair follicle pathologies [131]. Activated NF-κβ in dermal fibroblasts further stimulate infiltration of inflammatory cells, such as neutrophils, by stimulation of proinflammatory IL-1, IL-6, VEGF and TNF-α production. These cytokines stimulate neutrophils to release neutrophil collagenase (MMP 8), leading to matrix degradation and accelerated skin aging in the irradiated zones [132].

ROS may further damage the skin by stimulating the synthesis of proteolytic matrix metalloproteinases (MMPs) via MAPKs induction. Together, MMPs can fully degrade collagen [133], thus decreasing the skin elasticity. To maintain the collagen fiber content in the skin, the tissue-specific inhibitor (TIMP1) is essential to inhibit MMPs [134], especially MMP-1 (collagenase). Loss of balance between TIMP1 and MMPs can contribute to wrinkle development [131,134]. In addition, granzyme B-knockout mice showed decreased wrinkle formation after chronic UV exposure [135], which has suggested that inhibitory regulation of MMP-1, granzyme B and other PG-degrading proteases may serve as one of the anti-aging target mechanisms.

ROS may exert harmful effects by interfering with the nuclear factor erythroid 2-like 2 (Nrf2) that is a master regulator of the antioxidant responses. The Nrf2 is crucial to activate the antioxidant system and prevent further generation of ROS in all cell types of the skin. Many cytoprotective proteins, including heme oxygenase (HO-1), peroxiredoxins, NAPD(H) dehydrogenase, quinone 1 (NQO1), and the glutathione biosynthesis enzymes are downstream of the Nrf2 [136]. Therefore, Nrf2 is a key transcription factor regulating redox balance in skin aging.

## 7. Anti-Aging Strategies

While aging as a natural phenomenon is genetically determined, premature photoaging can be prevented. Wrinkling and pigmentation are directly associated with premature skin aging and are considered to be the most critical skin events [137]. Photoprotection achieved by physical and chemical UV filters is the main preventive measure against skin photo-damage. Use of nutraceuticals (the term is derived from “nutrition” and “pharmaceutical” [138]) represent a promising strategy for preventing, delaying or minimizing the premature skin aging and age-associated diseases, including skin cancers [139]. Among them are plant polyphenols, bioactive peptides and oligosaccharides, carotenoids, vitamins and polyunsaturated fatty acids. Although some studies have reported that polyphenols can exert cytotoxic effect, polyphenolic compounds (curcumin; polyphenols from green tee, grape, soybeans, pomegranate, etc.) belong to the most frequently used ingredients in modern cosmeceutical and dermatological products [125,140,141,142,143]. Numerous studies suggest that polyphenols modulate the cellular inflammatory response of the NF-κβ pathway [144,145] and exert indirect antioxidant actions via activation of the Nrf2 [146].

Topical nicotinamide (niacinamide, vitamin B3) improves skin appearance and provides beneficial effects in prevention of the loss of dermal collagen that characterizes photoaging [147,148,149]. Vitamin B_3_, a precursor of Nicotinamide Adenine Dinucleotide (NAD), can also prevent UV-induced depletion of ATP in keratinocytes, leading to the acceleration of energy-dependent DNA repair processes [150]. When DNA damage cannot be repaired, an activation of poly-ADP-ribose-polymerase (PARP-1) induces apoptosis by activation NF-κβ pathway [151]. Hence, the UV-protective effects of vitamin B_3_ on the skin include regulation of cellular metabolism [152,153]. The ability of nicotinamide to enhance PARP-1 and regulate DNA repair mechanisms lead to its inclusion in regular sunscreens [154,155].

The potent antioxidant properties of vitamins C and E are well known and documented. They are widely used for skin care and in photo-protection, either as nutraceuticals or for topical application [70]. The incorporation of ferulic acid improves chemical stability of the vitamins (C + E) and increases photo-protection of photo-exposed skin [156,157,158].

Another preventive measure against premature skin aging is the usage of vitamin D3 derivative_s_. It was reported that active forms of vitamin D3 protect, attenuate, or even reverse UVB-induced cell and DNA damage in skin cells [67,159,160,161,162,163,164,165]. Unfortunately, the chronic use of vitamin D3 at therapeutic doses in its classical active forms including 1,25(OH)_2_D_3_ is severely limited due to its calcemic (toxic) effects. However, the discovery of an alternative pathway of vitamin D activation initiated by CYP11A1 [36,37,38], which produces biologically active but non-calcemic novel derivatives detectable in vivo [166,167,168,169], offers promises for therapeutic applications against photoaging and UVR induced skin pathology [170]. Vitamin D analogs may increase the DNA repair capacity in keratinocytes and melanocytes by enhancement of the expression of tumor suppressor protein p53 phosphorylated at Ser-15, but not at Ser-46 [171]. Phosphorylation at Ser-15 and Ser-20 of p53 activates p53 and promotes DNA repair, with phosphorylation of p53 at Ser-46 being responsible for regulation of apoptosis after DNA damage [172]. In addition, novel vitamin D derivatives produced by CYP11A1 down-regulate the formation of mutagenic and genotoxic cyclobutane pyrimidine dimers (CPD) produced after UVB exposure. 

Thus, both classical 1,25(OH)_2_D_3_ [160,161] and novel CYP11A1-derived 20(OH)D_3_ and 20,23(OH)_2_D_3_, and other vitamin D3 derivatives, may work as protectors of the human epidermis against UV-induced oxidative damage, not only in keratinocytes but also in melanocytes [171].

Vitamin D3, production of which in the skin is induced by solar radiation, is essentially important as a protector of skin homeostasis [173]. It can attenuate DNA- and metabolic-damage by reducing H_2_O_2_ and NO levels, elevating glutathione levels, and enhancing DNA repair. In advanced age, the capacity of the skin to produce vitamin D, which could be a part of this intrinsic protective mechanism against UV-damage, declines. Therefore, the supplementation of vitamin D is of great importance in the elderly population.

The most promising candidate for delaying skin aging and for the treatment of several dermatoses associated with oxidative damage is melatonin. Melatonin is the main secretory hormonal product of the pineal gland and a regulator of chronobiological activities. Melatonin is also synthesized in numerous extrapineal sites including skin and hair follicles [54,58,174,175] where it can act on functional melatonin type 1 and 2 receptors (MT1 and MT2) [48,53,176,177,178,179,180]. Surprisingly, it was found that skin produces a much higher amount of melatonin for its own use than can be detected in serum [54,175]. Skin melatonin exerts multifaceted functions [179,180]. In addition to receptor-mediated actions, melatonin and its metabolites act as relevant direct antioxidants, as shown in Figure 3. Moreover, melatonin is one of the most potent free radical scavengers [181,182,183], even stronger than vitamins C and E [184]. Several in vitro studies have confirmed that melatonin and its metabolites can protect keratinocytes and melanocytes from UVB-induced damages. The mechanism of this protection includes activation of Nrf2 and upregulation of the Nrf2-related pathway [185,186]. Similarly, melatonin protects dermal fibroblasts from solar irradiation by increasing HO-1 expression and restoring the physiological expression of ECM proteins [187,188]. Melatonin reduces oxidative stress, not only as a direct ROS/RNS scavenger, but also indirectly via stimulation of antioxidant enzymes and inhibition of pro-oxidant enzymes [183,189]. Indeed, melatonin can upregulate expression of antioxidant genes [55,185,186,190]. Melatonin and its metabolites could also protect DNA from oxidative damages and reduce the levels of CPD’s or pyrimidine photoproducts (6-4PP) [185,191,192]. Melatonin, as an endogenous regulator, similarly to vitamin D3, stimulates phosphorylation of p53 at Ser-15 and enhances nucleotide excision repair (NER), thus preventing accumulation of damaged DNA and promoting antitumor activity [177,186,193].

Apart from its anti-oxidative properties, melatonin also preserves mitochondrial function. As we previously proposed, photoprotective functions of melatonin and its metabolites are directly or indirectly dependent on mitochondria, which appear to be a central hub of melatonin metabolism in skin cells [56]. Melatonin protects mitochondria not only directly, by ROS scavenging but also via maintenance of mitochondrial membrane potential and mitochondrial homeostasis in UV-exposed keratinocytes [56,194]. Additionally, melatonin and its metabolites ameliorate UVR-induced mitochondrial oxidative stress in human MNT-1 melanoma cells [195]. These data support the development of novel mitochondria-targeted antioxidants based on melatonin.

Furthermore, the lightening effects of melatonin and some of its metabolites are due to inhibition of proliferation and tyrosinase activity in epidermal melanocytes [175]. Since melatonin and its metabolites over the years have proved their cytoprotective and antiaging properties, topical application of exogenous melatonin and/or metabolites would be a useful strategy against skin aging [196,197].

To enhance the protective effects and prevent wrinkle formation during photoaging, sunscreens and antioxidants (topical and systemic including vitamin C) often are combined with retinoids. The use of retinoids can promote collagen production [137]. Retinoids, especially retinoic acids (RAs) enhance the steroidogenic potential in many classical and non-classical steroidogenic tissues, which decrease due to hormonal imbalance in aging [7,29,198]. Local regulation of steroidogenic activity in keratinocytes of the epidermis is important for skin physiology and homeostasis. RAs improve wrinkled appearance, post-inflammatory hyperpigmentation and inhibit differentiation of keratinocytes in both mice and humans [30], but they often lead to irritation.

## Figures and Tables

**Figure 1 ijms-20-02798-f001:**
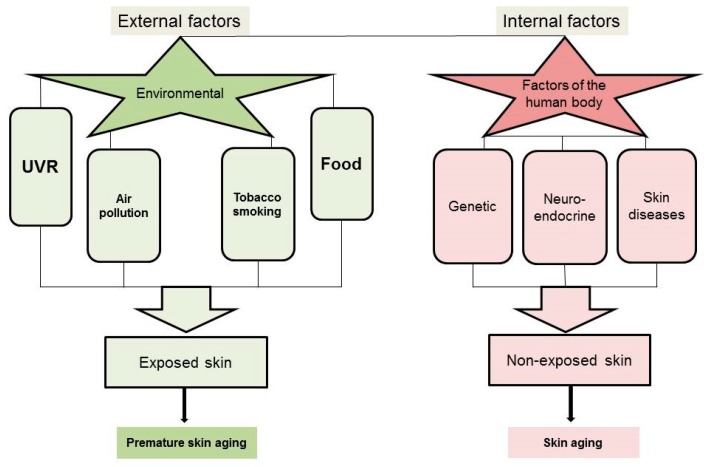
External and internal factors affecting skin aging.

**Figure 2 ijms-20-02798-f002:**
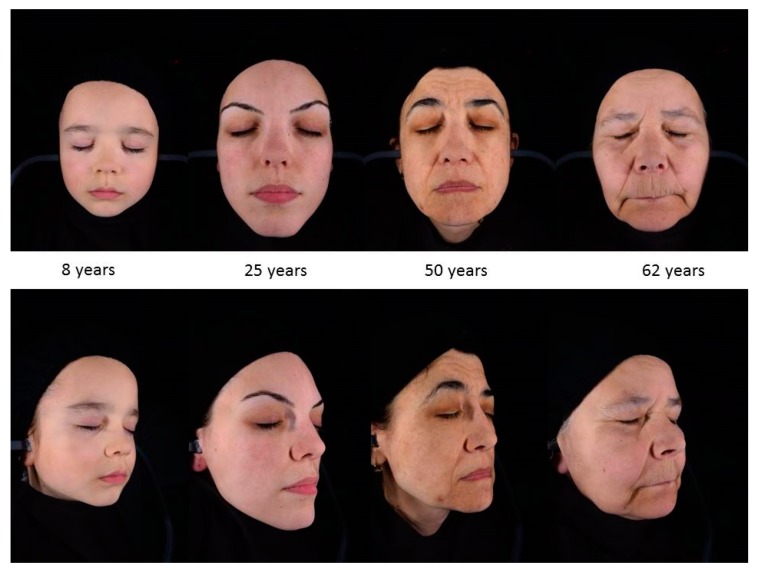
Chronology of aging. A written informed consent for publication has been obtained from participating volunteers.

**Figure 3 ijms-20-02798-f003:**
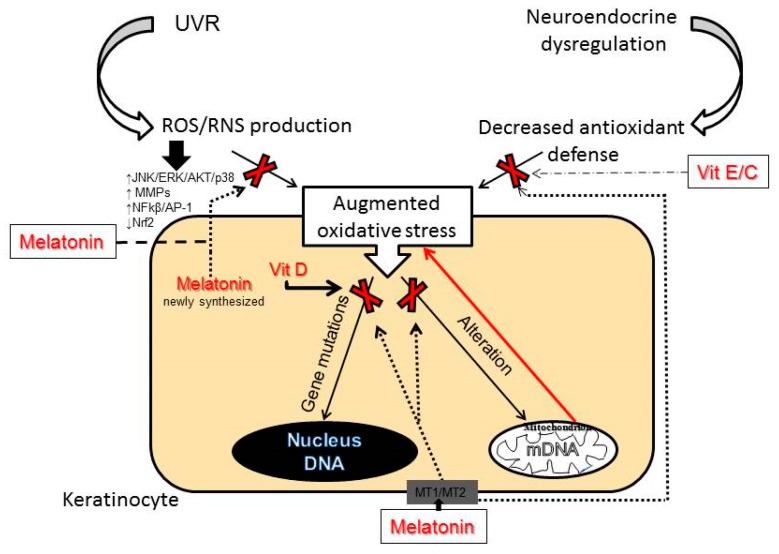
A role for melatonin, vitamins D3, E and C in the prevention and the treatment of oxidative stress-induced skin aging.

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
