# Peer review of "Neuroendocrine Aspects of Skin Aging"

_ijms, 2019, doi:10.3390/ijms20112798_

Round 1
Reviewer 1 Report
The authors have made a great effort to include the available information on skin aging and particularly on the neuroendocrine aspects involved in such process. I consider this a very complete revisionConcerning UVR, I would add concrete information on the influence of the diverse parameters that affect UV intensity (latitude -in which sense-, season -which-, time -which hours-). These are quite well-known/basic data, but I consider that they should appear in a revision like this one. It would be, if possible, very useful a scheme on point 4 (skin structure and morphological changes with age). I would add more information on retinoid acid, which is a well-established anti-aging product. Also, I would add a little more information on vitamin C. Probably, the text might be reduced so as to make it more easily readable. Be careful with spelling mistakes (e.g. page 2, line 76: where it says “glad” it should say “gland”). Congratulations for your great job.
Author Response
The manuscript entitled “Neuroendocrine aspects of skin aging” has been revised following reviewers suggestions and recommendations.
We greatly appreciate the reviewers’ critique that has allowed to improve the final presentation of the manuscript.
Answer to the reviewer 1:
We thank the Reviewer for his/her effort to improve our presentation. We revised the manuscript as requested.
As relates to details of UVR parameters we refer the reader to the citation 85 in the review (Marionnet, C et al., 2014, published in the same journal, IJMS) to avoid repetitions.
We agree that vitamin C is well known and important antioxidant that deserves a separate consideration for separate review or commentary. Following the reviewer’s critique, we have emphasized the role of vitamin C in photoprotection, see lines 403 and 404.
The role of retinoids is further discussed on lines 404 and 405 that is in addition to lines 405-407.
We have attempted to reduce the text as much as possible
Typographical errors have been corrected. Thank you very much for your attention.
Reviewer 2 Report
1. Institution 4 is missing (Slominski).
2. Section 2, Skin ageing : the concept of dermatoporosis, introduced in 2007, is missing (Kaya and Saurat, Dermatology 215:284-294, 2007; Kaya et al., J. Eur. Acad. Dermatol. Venereol. 32:189-191, 2018; Dyer & Miller, J. Clin. Aesthet. Dermatol. 11:13-18, 2018). This should be mentioned in a review on skin ageing.
3. Sentence lines 92-94 : The phrase "Putative mechanism of" can be cancelled; "environmentally induced aging" is known as "intrinsic aging".
4. Sentence lines 132-133 ("Outdoor workers ... ") : It can be mentioned that outdoor workers are at higher risk to develop squamous cell carcinoma, compared to indoor workers (because they accumulate a high total UV dose), but they are at lower risk to develop melanoma, compared to indoor workers who are exposed time to time to sunlight and develop sunburns (Gilchrest et al., N. Engl. J. Med. 340:1341-1348, 1999).
5. AT Slominski, the last author of this review, is author in 60 of 191 cited references ! This is clearly too much, in particular in a field as wide as skin ageing, even for its neuroendocrine part.
6. Sentence lines 168-169 ("Moreover, in vitro studies ...") : this may have implications only for sun radiations with wavelengths able to cross the epidermis and reach fibroblasts, i.e. > 310 nm for phototype I or > 340 nm for phototype V (Anderson and Parrish, J. Invest. Dermatol. 77:13-19, 1981; D'Orazio et al., Int. J. Mol. Sci. 14:12222-12248, 2013). This could be mentioned.
7. Lines 259-260 ("UVB radiation, absorbed mainly by epidermal DNA and RNA") : the first UVB chromophore encountered by UVB radiations when penetrating the skin is trans-urocanic acid, which prevents most of UVB radiations from reaching nucleic acids in the viable epidermal layers. Trans-urocanic acid endergoes a cis-transisomerisation to cis-urocanic acid, which is believed to mediate, at least in part, the UVB-induced immunosuppresion (Gibbs and Norval, J. Invest. Dermatol. 131:14-17, 2011). This is important to be indicated.
8. The sentence lines 393-395 ("To protective and ... with retinoids") is not clear; in particular, how can systemic antioxidants by included into sunscreens ?
Author Response
The manuscript entitled “Neuroendocrine aspects of skin aging” has been revised following reviewers suggestions and recommendations.
We greatly appreciate the reviewers’ critique that has allowed to improve the final presentation of the manuscript.
Answer to the reviewer 2:
1. Institution 4 is missing (Slominski).
Reply
This has been corrected
2. Section 2, Skin ageing : the concept of dermatoporosis, introduced in 2007, is missing (Kaya and Saurat, Dermatology 215:284-294, 2007; Kaya et al., J. Eur. Acad. Dermatol. Venereol. 32:189-191, 2018; Dyer & Miller, J. Clin. Aesthet. Dermatol. 11:13-18, 2018). This should be mentioned in a review on skin ageing.
Reply
This has been corrected, see line 207-209
3. Sentence lines 92-94 : The phrase "Putative mechanism of" can be cancelled; "environmentally induced aging" is known as "intrinsic aging".
Reply
This has been corrected
4. Sentence lines 132-133 ("Outdoor workers ... ") : It can be mentioned that outdoor workers are at higher risk to develop squamous cell carcinoma, compared to indoor workers (because they accumulate a high total UV dose), but they are at lower risk to develop melanoma, compared to indoor workers who are exposed time to time to sunlight and develop sunburns (Gilchrest et al., N. Engl. J. Med. 340:1341-1348, 1999).
Reply
This has been corrected, see revised lines 132-135.
5. AT Slominski, the last author of this review, is author in 60 of 191 cited references! This is clearly too much, in particular in a field as wide as skin ageing, even for its neuroendocrine part.
Reply
We attempted to reduce number of citation as possible and cited several reviews instead of original papers. These papers are in high impact factor journals and represent authoritative reviews on the subject. However, we increase the number of other citations requested by the reviewer.
6. Sentence lines 168-169 ("Moreover, in vitro studies ...") : this may have implications only for sun radiations with wavelengths able to cross the epidermis and reach fibroblasts, i.e. > 310 nm for phototype I or > 340 nm for phototype V (Anderson and Parrish, J. Invest. Dermatol. 77:13-19, 1981; D'Orazio et al., Int. J. Mol. Sci. 14:12222-12248, 2013). This could be mentioned.
Reply
This has been corrected, see lines 169-171
7. Lines 259-260 ("UVB radiation, absorbed mainly by epidermal DNA and RNA") : the first UVB chromophore encountered by UVB radiations when penetrating the skin is trans-urocanic acid, which prevents most of UVB radiations from reaching nucleic acids in the viable epidermal layers. Trans-urocanic acid endergoes a cis-transisomerisation to cis-urocanic acid, which is believed to mediate, at least in part, the UVB-induced immunosuppression (Gibbs and Norval, J. Invest. Dermatol. 131:14-17, 2011). This is important to be indicated.
Reply
This has been corrected, see lines 264-268.
8. The sentence lines 393-395 ("To protective and ... with retinoids") is not clear; in particular, how can systemic antioxidants by included into sunscreens ?
Reply
This has been corrected, see lines 403-405.
Additional changes
The correction in reply to reviewers “critique are marked in green. The correction of the text marked by editors in yellow are marked with red font.
The English has been corrected by the native English speaker